# RFC on RFCs

## Time Machine RFC-0000

Frédéric Kaplan      Kevin Baumer      Mike Kestemont

Daniel Jeller

## Motivation

Reaching consensus on the technology options to pursue in a programme as large as Time Machine is a complex issue. To ensure the open development and evaluation of work, a process inspired by the Request for Comments (RFC) that was used for the development of the Internet protocol[1] is being adapted to the needs of Time Machine. Time Machine Requests for Comments are freely accessible publications, identified with a unique ID, that constitute the main process for establishing rules, recommendations and core architectural choices for Time Machine components.

## Approach

The Time Machine RFCs are based on the following principles:

1. Accessibility. **RFCs** are freely accessible, at no cost.
2. Openness. Anybody can write an **RFC**.
3. Identification. Each **RFC**, once published, has a unique ID and version number. It can nevertheless be revised over time as a living document, being republished with the same ID and a different version number.
4. Incrementalism. Each **RFC** should be useful in its own right and act as a building block for others. Each **RFC** must be intended as a contribution, extension or revision of the Time Machine Infrastructure.
5. Standardisation. **RFCs** should aim to make use of standardised terms to improve the clarity level of its recommendation.
6. Scope. **RFCs** are designed contributions and implementation solutions for solving practical problems. **RFCs** are not research papers and may not necessarily contain experimental evidence. RFCs cover not only the technical infrastructure but the data standards, legal frameworks, and values and principles of Time Machine.

---

[1] https://tools.ietf.org/html/rfc791

7. Self-defining process. As used for the development of the Internet, **RFCs** are the main process for establishing Time Machine Infrastructure and Processes and also the processes and roles for managing **RFCs** themselves.

# RFC Publication Process

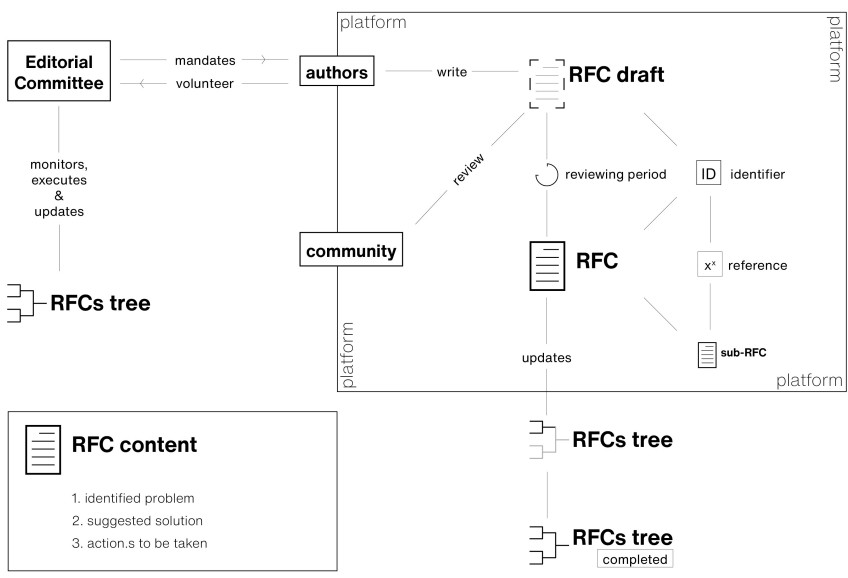

Figure 1: 75 % center

The **RFC Editorial Committee** organises the publication process of the RFCs, maintains the consistency of the RFC System, appoints RFC teams to organise new RFCs and to improve existing RFCs, keeps track of RFC versioning, ensures the timely and regular publication of RFCs, and is responsible for the public announcement of the open review process. The governance and organisation of the **RFC Editorial Committee** is defined in **RFC-0004**.

The publication process is the following :

1. The **RFC Editorial Committee** appoints authors to write the RFCs planned in the **RFC tree** (RFC-0002). Alternatively, authors may contact the **RFC Editorial Committee** to submit their candidature to write an RFC (planned in the **RFC tree** or not).
2. The authors produce an RFC draft which is reviewed, first by the **RFC Editorial Committee** for coherence with the rest of the RFC corpus and then by a larger community. The RFC is revised and possibly sent for review again.

3. Once accepted by the **RFC Editorial Committee**, an RFC receives an official identifier and is officially published as an peer-reviewed publication with proper scholarly credits assigned to the original author(s).
4. The **RFC tree** is adapted to include the published RFC and any possible sub-RFCs planned during the writing of the RFC.

## RFC Format

The RFC Format and Guidelines are estabished iteratively by the **RFC Editorial Committee**. The most-up-to-date version can be found in the **RFC-0000**.

Current Format

1. Motivation section
2. Series of sections describing the Approach and Solution
3. Question and Answers section
4. Linked RFCs section

## Question and Answers

### What are the main differences between Time Machine RFCs and Internet Society RFCs?

The Time Machine RFCs are being developed over 50 years after the RFCs that shaped in the Internet. The main differences are the following:

1. Time Machine RFCs are exclusively used to describe motivated solutions and not general communication.
2. Time Machine RFCs can be revised and are redefined iteratively, wheraeas significant improvement on an Internet Society RFC led to the creation of new RFC.

## Linked RFCs

- The **RFC Tree** is kept up to date in **RFC-0002**.
- The details of the RFC platform are defined in the **RFC-0003**.
- The governance and function of the **RFC Editorial Committee** is defined in **RFC-0004**.

