# OpenReview forum: "RFC-0000 - RFC on RFCs"
_TimeMachine.eu/RFC_

### Official Review · ~Caroline_Maximoff1 · 2020-09-08
**good introduction into the topic, background and goals!**

**Confidence:** 4
**Rating:** 7

**Review:**

As a lay person who is absolutely new to the RFC-world :-P, the document provides a good basis explaining the why and how of the RFC and the idea behind it. I guess everything get's clearer as soon as there is an RFC on a more specific topic, not on the RFC itself :-)

Of course the reader (at the moment) lacks the actual RFC-0002 - 0004 which should be available at the same time in order to back up the explanations/quotes in the document.